# Mimicry and law: Experiments in a natural setting of a law company

**Wojciech Kulesza**[1], **Paweł Muniak**[1]*, **Dariusz Dolinski**[2], **Tomasz Antoszek**[3], **Agnieszka Świderska-Cieśla**[4], **Anna Nowak**[3], **Bernadetta Kowalczyk**[5]

1 SWPS University of Social Sciences and Humanities, Faculty of Psychology in Warsaw, Centre for Research on Social Relations, Warsaw, Masovia, Poland, 2 SWPS University of Social Sciences and Humanities, Faculty of Psychology in Wroclaw, Wrocław, Lower Silesia, Poland, 3 SWPS University of Social Sciences and Humanities, Faculty of Psychology and Law in Poznan, Poznań, Greater Poland, Poland, 4 Wyższa Szkoła Umiejętności Społecznych im. prof. Michała Iwaszkiewicza w Poznaniu, Poznań, Greater Poland, Poland, 5 SWPS University of Social Sciences and Humanities, Faculty of Psychology in Warsaw, Warsaw, Masovia, Poland

* pmuniak@swps.edu.pl

**Data Availability Statement:** Databases, and statistical analysis for the experiments are available

## Abstract

This paper joins an effort to build a relational approach to law practice by testing mimicry as a vehicle for building trust in a legal context. Mimicry research indicates that this phenomenon leads to benefits, like greater trust, willingness to help, and satisfaction from interactions, which shows a potential for practical applications in, for example, a legal context. In two experiments conducted in the natural setting of a legal company, the tendency to trust the mimicker on a yet unresearched and deep level, namely putting one's legal future and security in the hands of an attorney, was measured. Both experiments consistently showed a greater tendency to give legal representation to an attorney when she verbally (Experiment #1) as well as both verbally and nonverbally (Experiment #2) mimicked the client. This paper explores the potential of applying mimicry in a legal service environment, focusing on fostering cooperation in professional conversations. Furthermore, the study contributes to the existing literature on mimicry by examining its effect on trust. Possible dangers, future studies and limitations are also discussed.

## Introduction

In recent decades, research has shown that the understanding of the lawyer's role in the legal process has changed significantly. Previously it was postulated that, mainly, attorneys should gain a broad understanding of legal issues, and then learn how to disseminate it professionally (for example, while interacting with clients). More recently, it has been stressed that the ability to engage in and successfully maintain social relations with clients and other legal representatives is not any less important than the above mentioned points [1–3].

In the present paper we test a new tool for enhancing social relations in the context of law interactions, i.e., mimicry, as it is hypothesized to be a powerful mechanism responsible for starting and maintaining social relationships [4, 5].

at the Open Science Framework (OSF; https://osf.io/63yaf/).

**Funding:** (1). The publication co-financed by the Ministry of Education and Science subsidies for maintaining and developing the didactic and research potential of the SWPS University of Social Sciences and Humanities (number: SUB/INTERDYSC/2019/12). (2). SWPS University of Social Sciences and Humanities - internal grant from FRBN programme (Fundusz Rozwoju Badań Naukowych - Research Development Fund). The funders had no role in study design, data collection and analysis, decision to publish, or preparation of the manuscript.

**Competing interests:** The authors have declared that no competing interests exist.

## Building trust in a legal context

Research run in a legal context has mostly neglected the aspect of trust reporting, for example, power dynamics in the attorney-client relationship [6] as well as conflicts [7], while Howieson [8] showed that clients followed advice provided by their attorney when s/he provided status recognition, voice, benevolence and—importantly—trust. On the one hand, Grisso [9] indicated that many adolescents do not trust attorneys while Walker [10] showed that 30% of juveniles reported that they were afraid to tell their attorney certain information, which led to a situation of resistance in adequately assisting the attorney in their defense [9–11]. From this perspective, researching methods to establish trust in a legal context is crucial since trust is the key element in establishing a deep collaboration between the attorney and the client [12–14]. Without such a connection, outcomes of such collaborations may be severe and disappointing. For example, it has been shown [15] that inmates with a low level of trust in their attorney received a lengthy sentence after going to the trial. In contrast, high trust led to satisfaction with sentences provided by the court and attorney collaboration. In addition, findings suggest that perceptions about attorneys' interpersonal skills are as important as perceptions about legal skills in forming opinions about overall lawyering ability. It is recommended that attorneys employ a well-rounded assortment of inter-personal skills to foster their clients' trust and to make better use of the limited amount of time they have to spend with clients. Pierce and Brodsky [16] also showed that 163 convicted male juvenile offenders were less trusting of their attorneys when combined with misunderstandings about the role of defense counsel as well as court-related knowledge. It was found that court-related misunderstandings were associated with mistrust in attorneys showing that there is much potential in researching trust in a legal context.

## Mimicry

Mimicry is one of the most common social behaviors. Birds mimic each other while forming a "V" flock to save energy for long distance flights [17]. Fish also mimic the behaviors of others and form synchronous schools to reduce the risk of being harmed by a predator [18]. Additionally, humans perform mimicry from the very first hours after being born [19]. Later in life we mimic each other to be accepted [20], liked [4, 21, 22], helped [23–25], and trusted [26, 27]. Since mimicry is such a common behavior, it is postulated that mimicry is responsible for creating and maintaining social relations; it acts as a "social glue" [5, 28] among us making this process one of the most fundamental, among many others. In the present paper we propose a relatively new approach towards the mimicry phenomena: we measure the tendency to trust in a situation when one's future is placed in the hands of another, namely an attorney, in some of the most important aspects of our lives: future, property, and even life.

## Mimicry and possible implementation in a legal context

To the best of our knowledge no research on mimicry has been conducted in the legal context per se and the present paper fills this gap. However, the body of literature on this phenomenon provides several results hinting at an application in this specific setting: it increases persuasiveness, negotiation outcomes, rapport, trust, and sales. Before we outline our experiments run in a law company, let us review the existing literature on this issue.

Firstly, one of the most important aspects of almost every interaction in a legal context is persuasiveness [29–31]. Mimicry research supports this important aspect of work for every attorney by showing that this process leads to an increased perception of persuasiveness of the mimicker. In an experiment tackling this very issue, participants were individually tasked with making a decision on some issue, and later discussing it in a group of three [32]. The new pair of interlocutors was constructed by the confederates (collaborator of the experimenters) both

of whom–during a discussion–mimicked (or not) the participant's gestures. Interestingly, the first mimicking confederate agreed with the participant, whilst the second mimicking confederate did not. It was discovered that a person who was in agreement and mimicked was perceived as more persuasive.

Secondly, attorneys are often included in teams involved in negotiations, and trust is a very important aspect in every–not only legal–relationship during negotiations [2, 33]. Trust between an attorney and a client is a fundamental factor in every legal cooperation [34,35]. On the other hand, lack of trust leads not only to worse or even discontinued cooperation, it may also lead to an attorney feeling frustrated, heightened dissatisfaction from work, or even professional burnout [36]. Mimicry within this area also supports increasing outcomes within this domain. In two experiments, a dyad negotiated a job contract [37]. One person was assigned to the role of a recruiter, and the other to that of an applicant. The recruiter mimicked (or not) the applicant's words. Results from both experiments were consistent: both interlocutors benefitted–in terms of the negotiation outcomes–from mimicry.

In the next line of research, the same scenario of recruitment was employed [38]. This time mimicry took place on a nonverbal level, and more complex patterns were employed. In the first two conditions one person always imitated the other, whilst in the control condition no mimicry was performed. This time mimickers and cumulatively a pair benefitted (not the mimickee) from negotiations. In the next experiment, interlocutors negotiated the sale of a gas station. The simulation was complex since, at first glance, it was impossible to reach an agreement. On the one hand, the seller could not sell for less than $ 553,000. On the other hand, the buyer could not offer more than $ 500,000. On closer inspection, there was however a way out from this clinch, and it was tested if mimicry would lead to finding this solution: the seller could sell for less than the asking price if an offer of employment at the gas station after finalizing the deal was made. This option was also interesting for the buyer who would potentially keep experienced workers hired. The question was if this information—leading to a better if not only positive negotiation outcome—would be exchanged by the mimicked seller. It was hypothesized that mimicry might enable this process.

Another crucial disclosure for the negotiation outcome was why the seller planned to get rid of her/his business: burnout, the need to recuperate vital strength, and finally return to work. Again, it was expected that mimicry might elicit a tendency to share this intimate information. The results confirmed that the expectations for mimicry were correct: in most pairs in which mimicry was employed in negotiations, two-thirds of pairs reached an agreement through deepened, and truthful conversations.

In more recent work on mimicry and negotiations, rapport as a key mechanism for the aforementioned link was tested [39]. Interacting participants were instructed to either verbally mimic each other, for only one participant to mimic the other, or in the control condition no verbal mimicry took place. Consistent with previous works, negotiating pairs reached higher gains individually and jointly, and rapport was discovered to be a key component.

Thirdly, in the context of attorney-client and attorney-attorney communication, rapport is also a key component [40, 41]. Additionally, in this domain, research validates mimicry's usefulness in a legal context. On the grounds of clinical psychology it was shown that nonverbal mimicry is crucial for emotional rapport [42, 43], and for a better understanding [44]. Recently it was shown that mimicry is linked with therapeutic success [45] by reducing the severity of the mechanisms/causes that led to therapeutic intervention [46]. Taken together, on the one hand, it may be concluded that mimicry is a key mechanism crucial for rapport between interacting partners, and, on the other hand, in attorney-client/attorney interaction, rapport is also crucial since the interaction often touches on very sensitive aspects of one's life (and in the light of the previously discussed point: are shared more eagerly).

Fourthly, mimicry was shown to be an effective mechanism responsible for increasing the mimickee's willingness to buy products presented by the mimicker. It was shown in a beauty shop where mimicked (or not) patrons bought twice as much as in the no-mimicry condition [47]. In a store selling electronic devices, clients were more eager to buy a product presented by the mimicker [48]. From this perspective it would follow that a mimicking attorney would make the client more eager to accept proposed strategies/options, etc., and to pay more for the provided legal services.

Fifthly, mimicry influences justice-related judgments: it decreases mimickees' tragic tendency to place the blame on victims of violent crimes [49], as well as makes mimickees more eager to perceive the world as a just place [50].

Sixthly, it was shown that mimickees are more eager to provide not only direct help to a mimicker, for example, picking up items dropped by a mimicker [23], but also to others via charity donations (mimickers asked mimickees to financially support charity goals [24]). In the context of legal services, such aspects are difficult to over appreciate. It is crucial for an attorney to make the client collaborate on legal issues. As a result of mimicry, increased helping/prosocial tendencies are probably the best vehicles for such collaboration.

Finally, mimicry research also brings about negative consequences for the mimicker. Since rapport and higher/better negotiation outcomes stem from performed mimicry (e.g., by the attorney) it might encourage a massive tendency to incorporate this mechanism into every legal interaction. One should, however, keep in mind that mimicry may backfire in situations where truthfulness is crucial: when gathering information and testimonies. How might mimicry harm the mimicker/attorney? On the one hand, mimicry increases the chance of being lied to by the mimicker [22], while on the other hand, the mimicker (e.g., attorney) is less capable of detecting a lie delivered by the mimickee (client) in comparison to when an attorney does not mimic [50].

## Goal of the paper

Taken together, it is clear that the existing body of research on mimicry offers a great potential for implementation in a legal context, mainly for increased trust during professional interactions (e.g., attorney-client). It is widely expected that this especially important aspect of a relational approach to law practice–concentrated on building relationships as a key component of legal practice [2, 3] –may benefit from mimicry. The present work aims to test whether mimicry is beneficial in a law practice setting, and if it leads to heightened trust, which is a key component in relations with attorneys [34,35]. The biggest obstacle to understanding whether mimicry is beneficial in a legal context is that no research has been conducted in a natural setting (almost all the above works were run as simulations). It is especially important to run an experiment in a natural setting for, at least, two reasons:

(1) Researching actual/real behavior cannot be substituted by a simpler, and easier experimental paradigm in which participants only imagine that they are mimicking someone or being mimicked. Existing literature on this issue shows that imagining mimicry does not provide the same pattern of results as real perceptions of mimicry [21].

(2) Since the actual behavior of a specific person is highly related to the social surrounding/ situation in which this person finds themselves, we decided to run experiments directly in law companies where attorney-client interactions take place.

This caveat is addressed in the present paper by running two experiments in an actual law company.

The second caveat stemming from the above review is that, of course, while prosocial behaviors performed by the client/mimickee and trust are crucial in the legal context, picking

up items and donating small sums of money to a charity box, and testing trust in simulations (compared to real life situations) are far from real legal situations. In stark contrast, the tendency to trust one's attorney while making important decisions, such as during a case that can affect the course of one's life (imprisonment) or losing custody of a child and property, can be significant. In two experiments introduced below we also address this caveat by measuring the tendency to hire an (mimicking or not) attorney for a legal case.

Databases, and statistical analysis for the experiments are available at the Open Science Framework (OSF; https://osf.io/63yaf/).

## Methods

### Participants

We conducted two experiments. In both experiments the study covered clients of an individual law company. At the first of them—60 clients (32 women, 28 men), aged 18–81 years ($M_{age}$ = 41.95, $SD_{age}$ = 15.26) participated in the study. To become a participant, each client had to be alone (since the presence of others or observers changes the behavior of the target person; [51]) and had not been a client of the law company previously. Participants were randomly assigned to one of two between-subject conditions: verbal mimicry ($n$ = 30; females = 15, males = 15), and no mimicry ($n$ = 30; females = 17, males = 13). A preliminary statistical analysis demonstrated that the gender distribution across experimental conditions was equal ($p$ = .796). Participants did not receive any payment for participation. No data were excluded from the analysis.

We tested a sample of $N$ = 60 participants/clients due to limitations in recruiting a larger number of lawyers and clients who were willing to participate. As suggested by Lakens and Evers [52], we aimed to conduct a rigorous study within the practical limitations that we faced and made every effort to detect the smallest possible effect size. The sensitivity power analysis in G*Power [53] yielded that with 60 participants/clients and $df$ = 1, the smallest effect size we could detect at 80% power ($\alpha$ = .05) would be Cohen's $\omega$ = .36 (which is considered as medium effect size; [54]). Although our sample size was relatively small, we strongly believe that our results provide valuable insights into the topic under investigation, carrying important practical applications.

In the second experiment—90 clients (45 women, 45 men), aged 18–82 years ($M_{age}$ = 40.53, $SD_{age}$ = 13.60) participated in the study. Each client who entered the office became a participant. Participants were randomly assigned to one of three between-subject conditions: mixed mimicry ($n$ = 30; females = 15, males = 15), verbal mimicry ($n$ = 30; females = 15, males = 15), and no mimicry ($n$ = 30; females = 15, males = 15). A preliminary statistical analysis demonstrated that the gender distribution across experimental conditions was equal ($p$ = .999). Participants did not receive any payment for participation. No data were excluded from the analysis.

We tested a sample of $N$ = 90 participants/clients due to limitations in recruiting a larger number of willing lawyers and clients, and we made every effort to detect the smallest possible effect size following Lakens and Evers [52]. Using a sensitivity power analysis in G*Power [53], we found that with 90 participants/clients and $df$ = 2, the smallest effect size we could detect at 80% power ($\alpha$ = .05) would be Cohen's $\omega$ = .33 (which is considered a medium effect size; [54]).

### Procedure

Two experiments were conducted: The experimental paradigm was very similar for both experiments. The first experiment's study design was as follows: After entering the office, the

secretary had the first contact with the participant/client. She greeted the arriving person, set the purpose of the meeting, and then, by phone, announced the experimenter/lawyer located nearby. After that, the experimenter/attorney (female and in her thirties) greeted the participant/client and then invited the participant/client to the office. During the meeting, the verbal mimicry manipulation took place. Following the recommendation by Kulesza and colleagues [24], in the mimicry condition, participants were verbally mimicked (or not, in the control condition) by the experimenter. The experimenter/lawyer copied the statements, paraphrased, and adopted the participant's tone of voice. In the non-mimicry condition, the experimenter/lawyer replied with statements like "okay." The interaction during which participants/clients were mimicked (or not) lasted for 30 minutes of the meeting. Declaring whether the experimenter/lawyer would receive power of attorney served as the dependent variable (yes/no). To control further details of the experiment, we coded whether the respondent/client granted the experimenter/lawyer power of attorney in the case (civil or criminal) after the debriefing took place.

The second experiment's study design was very similar to the first, but with an essential change. We added a new experimental condition in which the experimenter/lawyer simultaneously incorporated verbal and non-verbal mimicry into their discussion with the client. In other words, in this new condition, the experimenter/lawyer simultaneously copied the statements, paraphrased, and adopted the participant's tone of voice, as well as copied behaviors just after the participant completed them. Such behaviors included hand gestures (e.g., placing them on or removing them from the counter) and arm gestures as well as body stance, such as leaning forward or sitting up straight.

## Ethical approvals

Due to the nature of the study, the experimenters had access to the personal data of the participants. However, the study adhered to all confidentiality requirements, in line with legal practices. This study was reviewed and approved by an institutional ethics committee [SWPS University of Social Sciences and Humanities in Wroclaw, Poland. Opinion number: 08/P/03/2020] before it began (data for this study were collected in 2021). Informed consent was obtained from all individual participants after the study (which is a standard procedure in field studies where naturalistic settings are also necessary), and the consent process was conducted verbally by the experimenters who also witnessed the consent process. After the experiment none of the participants declared discomfort, all agreed to the use of the gathered data, and did not report needing any further assistance. It is a standard procurement procedure in field studies and is recommended, for example, by Grzyb and Dolinski [55].

## Results

### Experiment 1

In the case of the first experiment we carried out a 2 (mimicry: verbal of mimicry vs. no mimicry) x 2 (power of attorney: yes vs. no) likelihood ratio test for the effect of mimicry on probability of giving the power of attorney to the mimicking lawyer. JASP statistical software [56] was used to conduct the analysis with a combination of R programming language implemented in RStudio [57] with a "ggstatsplot", "rstatix" and "rcompanion" packages [58–60].

A $\chi^2$ test between verbal mimicry and no mimicry conditions and the delegation of the power of attorney (yes/no) was significant, $\chi^2(1) = 4.27$, $p = .039$, Cohen's $\omega = .30$, $OR = 3.45$, 95% CI [1.19, 9.99], $BF_{01} = 0.23$; which can be considered substantial evidence for $H_1$. For more details, please see **Fig 1**.

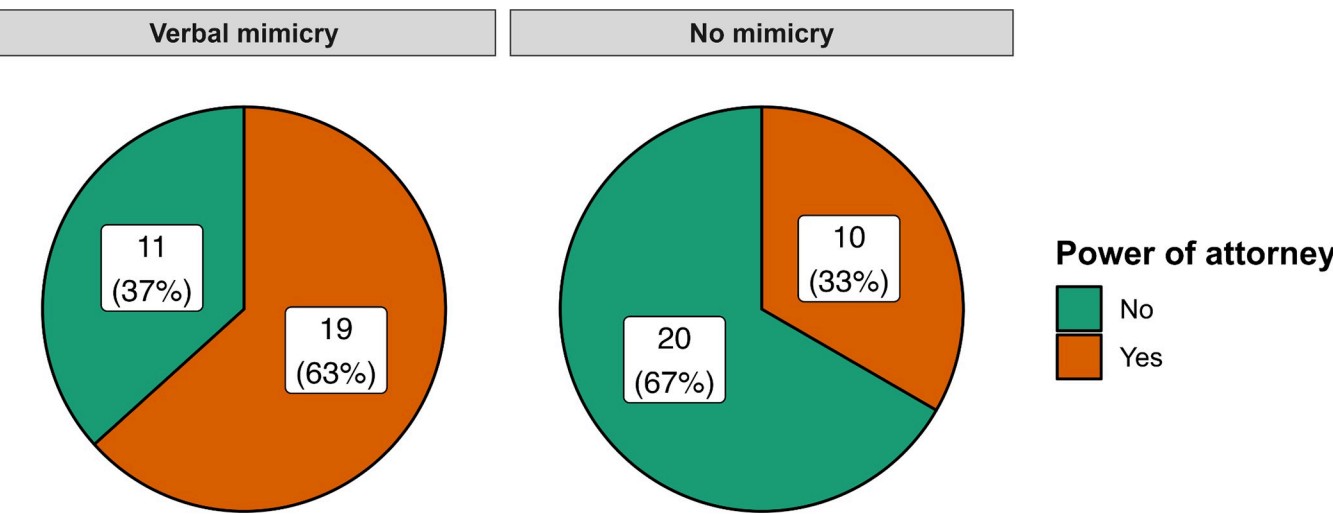

**Fig 1. Pie charts representing distribution of participants delegating the power of attorney in particular conditions in Experiment 1.**

Additionally, we ran a binominal test among those participants who delegated the power of attorney and who were mimicked, in order to test whether there were any differences considering the case type (civil/criminal). This analysis yielded no differences ($p$ = .999, $BF_{01}$ = 3.52; which can be considered substantial evidence for $H_0$).

Taken together, the results of the first experiment means that mimicked respondents/clients significantly often decided to give power of attorney to the experimenter/lawyer who was mimicking them regardless of the legal matter that was discussed. In other words, the effect of the mimicry manipulation during the meeting was effective in both civil and criminal cases.

It should be noted that in our study, the sensitivity power analysis yielded the smallest detectable effect size Cohen's $\omega$ = .36. However, ultimately, we obtained a lower effect size of Cohen's $\omega$ = .30. This indicates that our study did not have sufficient power. Therefore, it is critical to replicate this experiment with a larger sample to confirm the validity of our results and ensure they are not based on anecdotal data. Thus, the aim of the next experiment is to increase the sample size and test the generalizability of the effect of mimicry in a legal context. In the next experiment we also wanted to extend knowledge stemming from the first experiment, and to test if the combination of verbal and nonverbal mimicry (both described in the introduction; previously used by [61]) might increase (for a verbal mimicry condition) the tendency to grant legal representation (additive effect) or not.

### Experiment 2

In the case of the second experiment, we carried out a 3 (mimicry: mixed, verbal, no mimicry) x 2 (power of attorney: yes, no) $\chi^2$ test to test the effect of mimicry on probability of giving the power of attorney.

A $\chi^2$ test between mimicry conditions (mixed/verbal/no mimicry) and the delegation of the power of attorney (yes/no) was significant, $\chi^2(2)$ = 11.83, $p$ = .003, Cohen's $\omega$ = .36. When we compared the different types of mimicry using a post hoc test, we found that the difference in the number of participants who delegated power of attorney was significant between the mixed mimicry and no mimicry conditions ($p_{adj}$ = .005). The difference between the verbal mimicry and no mimicry conditions was not significant ($p_{adj}$ = .072). There was also no significant difference in the number of participants who delegated power of attorney between the mixed mimicry and verbal mimicry conditions ($p_{adj}$ = .422). For more details, please see **Fig 2**.

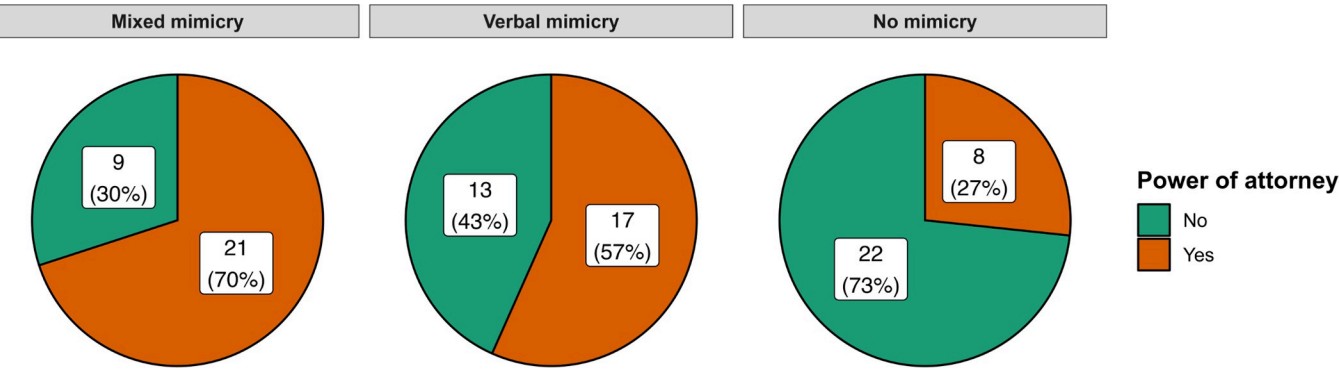

**Fig 2. Pie charts representing distribution of participants delegating the power of attorney in particular conditions in Experiment 2.**

Additionally, to gain more insights into the effect of mimicry (also in order to prepare data for a mini meta-analysis), we conducted two additional comparisons. Firstly, we compared the model with mixed mimicry and no mimicry, and then compared the model with verbal mimicry and no mimicry condition when considering the delegation of the power of attorney (yes/no). Both comparisons revealed a significant effect of mimicry (model with mixed mimicry: $\chi^2(1) = 9.61$, $p = .002$, Cohen's $\omega = .43$, $OR = 6.42$, 95% CI [1.86, 19.76], $BF_{01} = 0.01$; which can be considered very strong evidence for $H_1$ and the model with verbal mimicry: $\chi^2(1) = 4.39$, $p = .036$, Cohen's $\omega = .30$, $OR = 3.60$, 95% CI [1.22, 10.64], $BF_{01} = 0.21$; which can be considered substantial evidence for $H_1$).

Finally, we ran a binominal test among those participants who delegated the power of attorney and who were somehow mimicked, in order to test whether there were any differences considering the case type (civil/criminal). This analysis yielded no differences ($p = .999$, $BF_{01} = 5.02$; which can be considered substantial evidence for $H_0$).

Taken together, the second experiment replicates the findings from the first experiment. It also shows that combined mimicry also increases the chances of assigning power of attorney to the mimicking lawyer. In addition, we replicated that this effect is present in both civil and criminal cases. Although each of our two experiments provided valuable insights individually, we recognize that in both experiments, our relatively small sample size may have limited the statistical power and precision of effect estimates. To address this issue, we decided to conduct a mini meta-analysis by pooling the results across both studies [62]. By doing so, we aimed to enhance the statistical power, refine the effect estimates, strengthen the overall test of the effect, and improve the generalizability of the findings.

## Mini meta-analysis

Furthermore we carried out a random-effect meta-analysis, using REML estimation following Lipsey and Wilson [63] to investigate whether verbally mimicked participants/clients are more likely to give power of attorney to a mimicking lawyer (a mini meta-analysis in which we used a mixed mimicry condition, from Experiment 2, can be found in the **S1 Appendix**).

As our meta-analysis will be conducted using a small number of studies ($k = 2$) with small sample sizes (both $n = 60$), it is necessary to obtain stable and accurate estimates. As we report our results using $OR$ (odds ratios) in our mini meta-analysis, following Borenstein and colleagues' [64] recommendations, we will transform these values into $OR_{\text{logged}}$, i.e., the natural logarithm of $OR$. This will allow us to obtain more accurate and stable estimates of $OR$ and their standard errors, as well as make it easier to interpret the results.

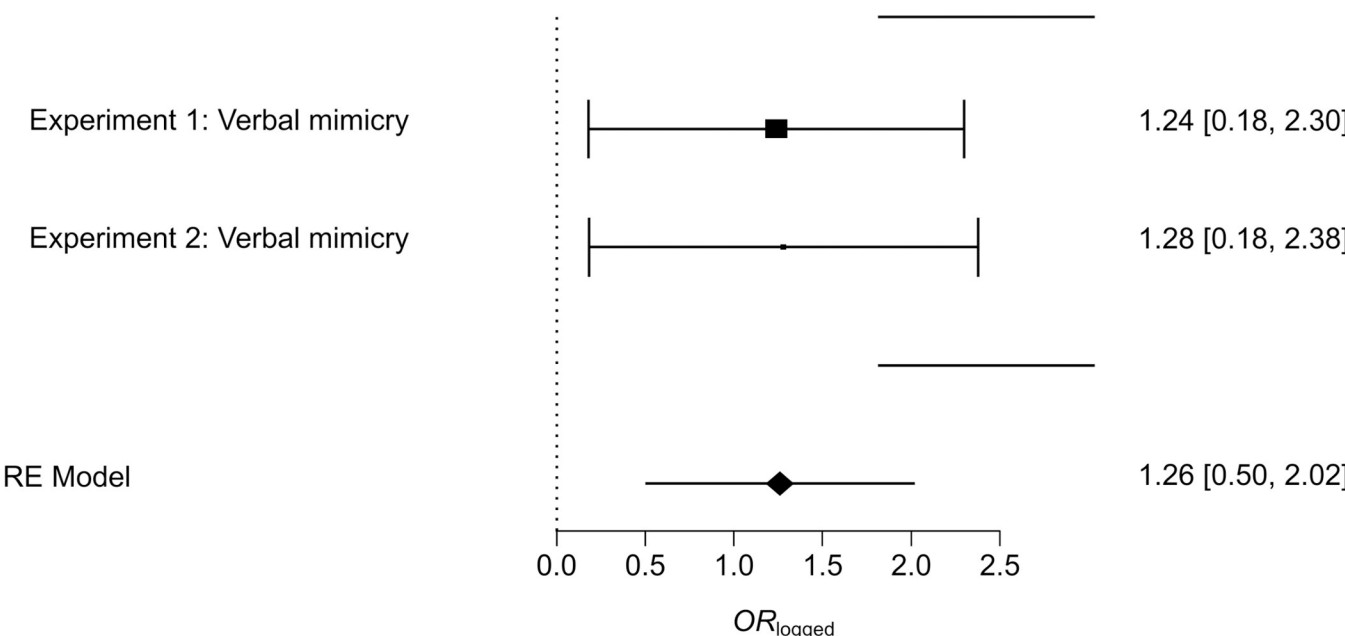

**Fig 3. Forest plot of meta-analyzed samples.**

The $OR_{logged}$ estimates ranged from *Min.* = 1.24 to *Max.* = 1.28. The estimated average $OR_{logged}$ based on the random-effects model was $OR_{logged}$ = 1.26, 95% CI [0.50, 2.02]. Therefore, the average outcome differed significantly from zero $z$ = 3.24, $p$ = .001. According to the $Q$-test, the true outcomes appear to be homogeneous, $Q(1)$ = 0.01, $p$ = .959, $\tau^2$ = 0.00, $I^2$ = 1%. A 95% prediction interval is given by 0.50 to 2.02. Hence, the average outcome is estimated to be positive, with the true outcomes being positive in both studies.

This mini meta-analysis indicates that a verbally mimicked participant/client has 3.53 ($OR$ = exp($OR_{logged}$) = exp(1.26) = 3.53) times greater odds of signing a power of attorney than a participant/client who was not mimicked. Furthermore, the 95% confidence interval for $OR$ [1.65, 7.54] does not contain the value of 1. This means that there is a high certainty that the relationship between verbally mimicking and signing a power of attorney is true and not due to chance. For more details, please see **Fig 3**.

## General discussion

From the perspective of research run in a legal context, this paper offers promising results. The vast majority of research has been run in laboratories, with a much smaller number run in natural (e.g., business charity) settings. No experiments had been run in the legal context (neither as simulations nor in natural settings) leaving a gap that is filled by the present paper. Following "a relational approach to law practice" concentrating on building relationships as a key component of legal practice [1–3], mimicry was shown to have a high potential for implementation in this special, and very important (for example, in terms of consequences of good collaborations between interacting dyads) practice.

In detail, two consecutive studies run in legal companies clearly show benefits for the attorney implementing mimicry during the very first interaction with a potential client: our participants granted permission for legal representation much more often in comparison to when the attorney did not implement mimicry. Importantly, in both conditions, attorneys were

professional, worked in the same environment, and provided the same level of expertise so mimicry is solely responsible for this outcome.

It was twice replicated that verbal mimicry is beneficial for the attorney since in the mimicry condition legal representation was granted significantly more often. The second experiment tested if the additive effect of two kinds of mimicry might increase this tendency to grant legal representation. However, we found that the increase was not significant.

Since trust is a foundation for every legal cooperation [34, 35], and lack thereof may be harmful to the attorney [36], our research clearly shows how to gain first (cooperation), and avoid second (harms): to mimic the client. This result clearly adds novelty to the body of literature on interpersonal skills that are important for the legal context [34, 65–67]. In addition, a study by Kim et al. [68] reports that certain personal skills that contribute to perceived mutual 'understanding' lead to greater trust and a stronger desire to renew cooperation in the future.

Also—besides legal aspects—for the psychological literature in general this paper offers conclusions. As mentioned above, helping and prosocial tendencies that stem from elicited trust have mainly been measured in laboratories, which are unnatural settings [23]. More importantly, dependent measures have usually been "trivial" like small monetary donations to the mimicker, or picking up items dropped by the mimicker. This research expands the literature by showing that mimicry creates the tendency to grant the mimicker trust on a much deeper level: like one's life, future, and property.

This supports—on a more general level—the existing body of literature on mimicry: the mimicry as a social glue hypothesis [5, 28], which states that mimicry is responsible for creating and maintaining social relationships. Our research shows how deeply involved in relationships and trust this process is.

## Practical implications

It is clear that building professional relationships, that is, a relational approach to law practice, is a key component of legal practice [1–3], and trust (or lack thereof) is highlighted as highly crucial in this process [34–36]. The present paper joins this effort by recommending future directions in which the results discussed here might be employed. There are several practical implications stemming from our research.

The first is very obvious: mimicry should be employed in legal contexts during interactions with clients. It is very possible that this phenomenon might be applied in other professional discussions between lawyers. However, the question remains unanswered as to whether mimicry is influential for mutual professional perceptions (given that legal representation is not applicable in such a context). In the next section we elaborate more on this issue.

Second, mimicry should be taught during legal studies and during practice for adepts of legal professions. For example, medical professions are very aware of benefits stemming from mimicry and have already started the process of dissemination among (future) adepts [69, 70]. The same path is recommended in the practice of law.

Third, although advertisements for legal services are forbidden in some countries, mimicry can lead to increases in sales [47, 48]. From this perspective our paper indicates the possibility to increase sales without advertisements. Direct meetings with different attorneys may lead to the conclusion that the mimicking one is the most trustworthy and thus will be hired for legal implementation; without advertising one's practice s/he can increase profits in comparison to other legal companies/attorneys.

Importantly, the picture of applying mimicry in the legal context—from the present paper as well as the existing body of literature—is complex. On the one hand, our experiments clearly show benefits for attorneys who apply mimicry in the legal context of client interactions. On

the other hand, one should keep in mind that mimicry brings risks and costs for the mimicker, such as a greater tendency to lie by the mimickee [22]. Mimickers also present a decreased ability to detect lies [50]. Additionally, while this context should be strengthened during dissemination of knowledge from our research, other results should be included to avoid implementation without careful consideration of possible costs.

It is possible that mimicry might only be very useful for an attorney applying this phenomenon in professional discussions with clients during first interactions. Once the legal representation is granted to the lawyer, and the next meeting takes place, s/he might refrain from mimicry to avoid the aforementioned costs. In other words, the costs of mimicry might possibly occur in later interactions when the details of the case are given to the attorney. It is unnatural to disseminate all—sometimes incriminating details—prior to legal representation making any future discussion confidential. However, this is pure speculation, and future studies should address this important distinction leading to gains in benefits from mimicry whilst avoiding costs. In the next section we elaborate on other ideas for future research that expand the results described in this paper.

Finally, since the combination of mimicry did not increase the tendency to grant legal representation to the attorney in comparison to solely verbal mimicry, it is clear that lawyers do not have to employ further mimicry. In other words, verbal mimicry produces the same result as more effortful interactions.

## Future directions

Since our research is first to be run in the context of legal services, it answers some research questions but, at the same time, it creates many more. Firstly, the temporal aspects of mimicry have recently been reported, answering the question of how long one should mimic to be granted benefits. It was shown that mimicry is beneficial even after a short period (5 minutes; and that mimicry outcomes were the highest when mimicry was applied twice (at the beginning and at the end of an interaction; [71, 72]).

Since the presented studies are new in the field of legal practice, it is hard to estimate at which point the benefits from the performed mimicry will disappear. In our research, the question about transferring (or not) legal representation to the lawyer was asked immediately after mimicry behavior ended. With the present data in hand, it is unknown when the legal representation transfer should take place: after interactions in which mimicry is employed only, or —for example—the next day. In general, it is unclear if the effect remains after the client leaves the office and decides at a later time. Would the impact of mimicry be strong enough to last for a longer time? In the same vein it would be interesting to assess how long the mimicry should be performed for it to produce the reported outcomes.

Secondly, it has been shown that the mimicry effect may transfer to other, previously non-mimicking people: participants donated money not only to the mimicker, but also to others, unrelated to the mimicker (but, importantly, only after the presence of mimicry [73]). Furthermore, research has shown that the effects of mimicry are even more far-reaching. A salesperson or receptionist who mimics another person is not only viewed more positively by the person being mimicked, but also positively influences the perception of the entire organization they represent [74].

In the context of a legal office, it would be interesting to see if the mimicry of one attorney may transfer to another (non-mimicking) lawyer. In this case mimicry might be a double-edged blade: in cases when the client meets another attorney in the same office, mimicry is beneficial; in cases when the client attends another meeting at a different company, mimicry would backfire for the first attorney. Pairing this study with the previous one (temporal aspects

of mimicry: how long it holds its influence) would produce clear recommendations in these manners.

Thirdly, as mentioned in the introduction, mimicry research has its roots in clinical psychology, where it has been shown that mimicry is responsible for rapport in the clinician-client relationship [42, 43]. Since rapport is crucial in negotiations where mimicry is employed [39] it would be interesting to replicate our studies by assessing the rapport elicited between a client and a mimicking (or not) lawyer. Such studies might address a very important question that we are—with the present data—unable to address: why is it that mimicry creates a greater tendency in clients to transfer legal representation to the lawyer? It is possible that the mechanism behind the results reported here is rapport, but with the current data we are unable to address this possibility.

Fourthly, it was shown that mimicry influences justice-based judgments: it decreases mimickees' tragic tendency to place the blame on victims of violent crimes [49], as well as increases mimickees' eagerness to perceive the world as a just place [75]. Although such an idea is very difficult to implement in the naturalistic setting of a legal firm, it would be interesting to see what effect mimicry has on the perpetrator of a violent crime. Would it lead to an increased tendency to empathize with the victim? If so, it might reduce the tendency to repeat similar crimes, or even crimes in general. This in turn might lead to improving access to justice [1].

Fifthly, it would be very interesting to run experiments on attorney-attorney "of a different branch" (prosecutor, judge, defendants) conversations in which mimicry would be employed. Since mimicry is so powerful, it may be beneficial in building trust [26, 27], prosocial tendencies [23–25], and cooperation between legal professionals. If so, mimicry might also be beneficial in this context.

Finally, from the perspective of our research it is already known that the aforementioned interesting lines of research may be, however, very difficult to implement since legal firms are first and foremost responsible for delivering good quality legal knowledge, and—very often—financial benefits to their partners. From this perspective, conducting such research in this setting may be seen as posing a risk of losing clients, or as impeding the delivery of high quality legal services.

## Limitations

The novelty of the research is, one the one hand, an advantage (showing new potential), but, on the other hand, a disadvantage: our data (however replicated) may be restricted to the place of research (two legal companies in the same country). More replications in different cultures are highly recommended.

Since the research attorney was female, subsequent research should eliminate the possibility that our manipulation of mimicry in a legal context is restricted only to female lawyers. One should, however, keep in mind that, to date, in the literature on mimicry there are no reported systematic effects for female mimickers only. In the same vein full manipulation and control of gender (for both: mimicker/attorney and participant) are recommended to test how exactly mimicry in legal settings is an efficient mechanism for eliciting trust.

Secondly, on the grounds of the second experiment, the question may arise: why was testing two kinds of mimicry not run in a clear manner with a 2 (nonverbal mimicry: yes/no) x 2 (nonverbal mimicry: yes/no) experimental design? We were limited by the difficulty of recruitment: since participants were clients who (1) entered alone (2) and for the first time (3), the time needed to complete an experiment increased by 33% (from two to three conditions). Increasing time by 50% was not possible in the company we collaborated with. It is highly desirable that future studies address this important caveat stemming from our study.

Next, there are ethical applied limitations to this study. For example, what if an untrustworthy legal representative employs mimicry to create trust? The answer to this very important question may be the same as to any social influence technique: to make both parties of the interaction aware of such techniques. Since mimicry is also an effective social influence technique [47] such publications as this one provide such knowledge.

Another important ethical question is: can a profession based on social trust employ social influence tools to manipulate their clients? The answer is yes when we reconstruct this question in the environment of psychotherapists. In this context, trust has been researched for at least several decades [22] showing the importance of employing mimicry in a clinical context. Without trust, a therapeutic alliance is difficult to reach. It seems that the legal context is not far from the psychological profession: without trust, good cooperation while dealing with legal issues is difficult to imagine.

Last but not least, during our studies, we encountered practical challenges in recruiting a larger number of lawyers/participants, which may have affected the power of our analyses. However, we strengthened our analyses by conducting a power analysis, using Bayesian statistics, doing post hoc tests, and performing a mini meta-analysis. All of these steps significantly increased the reliability of our findings. Thus, despite the limitations imposed by our sample size, we can be reasonably confident in interpreting our data.

## Supporting information

**S1 Appendix. Mini meta-analysis with mixed mimicry condition.**
(DOCX)

## Author Contributions

**Conceptualization:** Wojciech Kulesza.

**Data curation:** Agnieszka Świderska-Cieśla, Anna Nowak.

**Formal analysis:** Paweł Muniak.

**Funding acquisition:** Wojciech Kulesza, Tomasz Antoszek.

**Investigation:** Wojciech Kulesza, Tomasz Antoszek, Agnieszka Świderska-Cieśla, Anna Nowak, Bernadetta Kowalczyk.

**Methodology:** Wojciech Kulesza.

**Project administration:** Wojciech Kulesza, Tomasz Antoszek, Agnieszka Świderska-Cieśla, Anna Nowak, Bernadetta Kowalczyk.

**Resources:** Wojciech Kulesza, Tomasz Antoszek, Agnieszka Świderska-Cieśla, Anna Nowak.

**Supervision:** Wojciech Kulesza, Dariusz Dolinski, Tomasz Antoszek.

**Validation:** Wojciech Kulesza, Dariusz Dolinski, Bernadetta Kowalczyk.

**Visualization:** Paweł Muniak.

**Writing – original draft:** Wojciech Kulesza, Paweł Muniak, Dariusz Dolinski.

**Writing – review & editing:** Wojciech Kulesza, Paweł Muniak, Dariusz Dolinski, Bernadetta Kowalczyk.

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
