## [Decision Letter · Decision Letter 0]

18 Jul 2023

PONE-D-23-10321

Mimicry and law: Experiments in a natural setting of a law company

PLOS ONE

Dear Dr. Muniak,

Thank you for submitting your manuscript to PLOS ONE. After careful consideration, we feel that it has merit but does not fully meet PLOS ONE’s publication criteria as it currently stands. Therefore, we invite you to submit a revised version of the manuscript that addresses the points raised during the review process.

1. We suggest that you kindly consider reorganizing your manuscript and report using the 'IMRaD method', i.e. Introduction, Methods, Results , analysis and Discussion, before Limitations, etc. As it currently stands your manuscript includes multiple sections on procedures, results, and discussions. We feel reorganization of the manuscript will make for easier readability.

2.  Also, at the end of your Methods section please include a sub-section on "ethical approvals", including information on methods of obtaining informed consent from participants and methods used to maintain data and participant confidentiality.

3. In your Introduction kindly expand briefly on the 'literature review with regards to trust building in legal contexts.'

4. In your discussion  kindly address the potential 'ethical considerations regarding the practical application of the results of your study, and also expatiate or reflect briefly 'on the effects of lawyers using mimicry in their interactions with clients.'

5. Finally, in your abstract and discussion, and conclusions kindly refrain from to 'overinterpretation on the importance of the obtained results' from this study, as suggested by one reviewer.

We look forward to receiving your revised manuscript.

Kind regards,

Sylvester Chidi Chima, M.D., L.L.M, LLD.

Academic Editor

PLOS ONE

Journal Requirements: 

"The publication co-financed by the Ministry of Education and Science subsidies for maintaining and developing the didactic and research potential of the SWPS University of Social Sciences and Humanities (number: SUB/INTERDYSC/2019/12)."     

4. Please upload a new copy of Figures 1 and 2 as the detail is not clear. Please follow the link for more information: " ext-link-type="uri" xlink:type="simple">https://blogs.plos.org/plos/2019/06/looking-good-tips-for-creating-your-plos-figures-graphics/"
https://blogs.plos.org/plos/2019/06/looking-good-tips-for-creating-your-plos-figures-graphics

5. Please include captions for your Supporting Information files at the end of your manuscript, and update any in-text citations to match accordingly. Please see our Supporting Information guidelines for more information: http://journals.plos.org/plosone/s/supporting-information

Reviewers' comments:

Reviewer's Responses to Questions

**Comments to the Author**

1. Is the manuscript technically sound, and do the data support the conclusions?

Reviewer #1: Yes

Reviewer #2: Yes

2. Has the statistical analysis been performed appropriately and rigorously? 

Reviewer #1: Yes

Reviewer #2: Yes

3. Have the authors made all data underlying the findings in their manuscript fully available?

Reviewer #1: Yes

Reviewer #2: Yes

4. Is the manuscript presented in an intelligible fashion and written in standard English?

Reviewer #1: Yes

Reviewer #2: Yes

5. Review Comments to the Author

Reviewer #1: Many thanks for this fascinating contribution. Mimicry among lawyers is, arguably, not studied at all, while it can contribute to successful negotiation strategies and trust building. I'm surprised you're not publishing this study in a more law-oriented journal, but PLOS One, being a prime generalist journal, should fit just as well.

My only minor comment is that you should expand a bit the lit review with regard to trust building in legal contexts, as you focus on psychology mainly.

Reviewer #2: I liked the article for several reasons. First, it describes a series of studies carried out in natural conditions, which is increasingly rare in psychology. Secondly, although the research consists of field studies, it is well-prepared. The sample sizes are carefully calculated using the G*Power program, and the statistical analyses are conducted properly and sensibly. Both studies form a series, where minor errors made in the first study are fixed in the second. Furthermore, the authors also perform a mini meta-analysis to present the combined results of both experiments.

However, the article does have minor drawbacks that, in my opinion, require improvement. The authors tend to overinterpret the importance of the obtained results to some extent. In certain cases, certain adjectives seem redundant, such as in the abstract, where the phrase 'great potential' appears rather overly optimistic to me. Additionally, the authors allocate too little space to the ethical considerations regarding the practical application of their results. I feel that the article lacks reflection on the effects of lawyers using mimicry in their interactions with clients. Can a profession based on social trust employ social influence tools to manipulate their clients? It seems that this issue is not given sufficient attention in the article.

Despite the mentioned faults, I find the submitted article interesting and believe it is worth publishing in PlosOne after implementing the necessary corrections.

6. PLOS authors have the option to publish the peer review history of their article (what does this mean?). If published, this will include your full peer review and any attached files.

Reviewer #1: No

Reviewer #2: No

---

## [Author Response · Author response to Decision Letter 0]

29 Aug 2023

1. We suggest that you kindly consider reorganizing your manuscript and report using the 'IMRaD method', i.e. Introduction, Methods, Results , analysis and Discussion, before Limitations, etc. As it currently stands your manuscript includes multiple sections on procedures, results, and discussions. We feel reorganization of the manuscript will make for easier readability.

- Thank you for your valuable feedback. We appreciate your suggestion to reorganize the manuscript using the 'IMRaD' method. We understand the importance of a clear and logical structure for the readability of the manuscript.

In response to your comments, we have revised our manuscript to follow the 'IMRaD' structure, i.e., Introduction, Methods, Results, Analysis, and Discussion, before the Limitations section. We have consolidated the multiple sections on procedures, results, and discussions into these categories to provide a more streamlined and coherent presentation.

We believe these changes will significantly improve the readability and flow of our manuscript. Thank you again for your insightful suggestions. Please see p. 10 to p. 12.

2. Also, at the end of your Methods section please include a sub-section on "ethical approvals", including information on methods of obtaining informed consent from participants and methods used to maintain data and participant confidentiality.

- Thank you for your comment. In accordance with your recommendation, the ethics statement has now been moved to the Methods sub-section “ethical approvals”.

It has been removed from all other sections.

Please see p. 12 to familiarize yourself with the changes. 

3. In your Introduction kindly expand briefly on the 'literature review with regards to trust building in legal contexts.'

- Thank you so much for this suggestion. The previous version of the manuscript clearly missed this important deliberation.

In the present version of the manuscript we have added the literature review on trust creation in the legal context (see p.3-4).

Thanks to this great comment we have - in the discussion section - also briefly described the input of our research to the body of literature on trust in legal contexts (see p.18).

4. In your discussion kindly address the potential 'ethical considerations regarding the practical application of the results of your study, and also expatiate or reflect briefly 'on the effects of lawyers using mimicry in their interactions with clients.'

- Done! Please see p.23 – 24.

5. Finally, in your abstract and discussion, and conclusions kindly refrain from to 'overinterpretation on the importance of the obtained results' from this study, as suggested by one reviewer.

- You are absolutely right! We were too “excited” while describing the results. In the present version of the manuscript we discuss (both: abstract and discussion) the results in a more restrained way.

● A rebuttal letter that responds to each point raised by the academic editor and reviewer(s). You should upload this letter as a separate file labeled 'Response to Reviewers'.

● A marked-up copy of your manuscript that highlights changes made to the original version. You should upload this as a separate file labeled 'Revised Manuscript with Track Changes'.

● An unmarked version of your revised paper without tracked changes. You should upload this as a separate file labeled 'Manuscript'.

If applicable, we recommend that you deposit your laboratory protocols in protocols.io to enhance the reproducibility of your results. Protocols.io assigns your protocol its own identifier (DOI) so that it can be cited independently in the future. For instructions see: https://journals.plos.org/plosone/s/submission-guidelines#loc-laboratory-protocols. Additionally, PLOS ONE offers an option for publishing peer-reviewed Lab Protocol articles, which describe protocols hosted on protocols.io. Read more information on sharing protocols at https://plos.org/protocols?utm_medium=editorial-emailutm_source=authorlettersutm_campaign=protocols.

We look forward to receiving your revised manuscript.

Kind regards,

Sylvester Chidi Chima, M.D., L.L.M, LLD.

Academic Editor

PLOS ONE

Journal Requirements: 

- Thank you for your comment. We appreciate your guidance. We have thoroughly reviewed PLOS ONE's style requirements and have made sure that our manuscript, including the file naming, is in full compliance with them. We understand the importance of adhering to these standards for a smooth and efficient review process.

"The publication co-financed by the Ministry of Education and Science subsidies for maintaining and developing the didactic and research potential of the SWPS University of Social Sciences and Humanities (number: SUB/INTERDYSC/2019/12)." 

- Thank you for your attention to our financial disclosure. We would like to clarify that the funders had no role in the study design, data collection and analysis, decision to publish, or preparation of the manuscript. We also add this important information to the cover letter. We appreciate your diligence in ensuring the transparency of our work.

- The ethics statement has now been moved to the Methods sub-section “ethical approvals” in the manuscript. It has been removed from all other sections.

Please see p. 12 to familiarize yourself with the changes. 

4. Please upload a new copy of Figures 1 and 2 as the detail is not clear. Please follow the link for more information: https://blogs.plos.org/plos/2019/06/looking-good-tips-for-creating-your-plos-figures-graphics/" https://blogs.plos.org/plos/2019/06/looking-good-tips-for-creating-your-plos-figures-graphics

- Thank you for your valuable feedback! We have taken your comments into consideration and have updated our figures to enhance their clarity as per your suggestion. The revised figures are now included in the updated version of the manuscript.

5. Please include captions for your Supporting Information files at the end of your manuscript, and update any in-text citations to match accordingly. Please see our Supporting Information guidelines for more information: http://journals.plos.org/plosone/s/supporting-information

- We have carefully revised the description of our Supporting Information files in accordance with the guidelines provided. The captions for the Supporting Information files have been added at the end of our manuscript, and in-text citations have been updated to match accordingly. 

Please see p. 16 and p. 33.

- Thank you very much for the call to action in this regard. The reference list has been verified as indicated. It is correct and complete. The referenced documents contained in the manuscript are authoritative. In addition, we inform you that the changes that have been made include the order of citations, caused by the introduction of corrections corresponding to the reorganization of the manuscript and report using the "IMRaD" method.

Reviewers' comments:

Reviewer's Responses to Questions

Comments to the Author

1. Is the manuscript technically sound, and do the data support the conclusions?

Reviewer #1: Yes

Reviewer #2: Yes

2. Has the statistical analysis been performed appropriately and rigorously?

Reviewer #1: Yes

Reviewer #2: Yes

3. Have the authors made all data underlying the findings in their manuscript fully available?

Reviewer #1: Yes

Reviewer #2: Yes

4. Is the manuscript presented in an intelligible fashion and written in standard English?

Reviewer #1: Yes

Reviewer #2: Yes

5. Review Comments to the Author

Reviewer #1: Many thanks for this fascinating contribution. Mimicry among lawyers is, arguably, not studied at all, while it can contribute to successful negotiation strategies and trust building. I'm surprised you're not publishing this study in a more law-oriented journal, but PLOS One, being a prime generalist journal, should fit just as well.

My only minor comment is that you should expand a bit the lit review with regard to trust building in legal contexts, as you focus on psychology mainly.

- Done! Please see the corresponding reply above in this response letter (see p. 3-4)

Reviewer #2: I liked the article for several reasons. First, it describes a series of studies carried out in natural conditions, which is increasingly rare in psychology. Secondly, although the research consists of field studies, it is well-prepared. The sample sizes are carefully calculated using the G*Power program, and the statistical analyses are conducted properly and sensibly. Both studies form a series, where minor errors made in the first study are fixed in the second. Furthermore, the authors also perform a mini meta-analysis to present the combined results of both experiments.

However, the article does have minor drawbacks that, in my opinion, require improvement. The authors tend to overinterpret the importance of the obtained results to some extent. In certain cases, certain adjectives seem redundant, such as in the abstract, where the phrase 'great potential' appears rather overly optimistic to me. Additionally, the authors allocate too little space to the ethical considerations regarding the practical application of their results. I feel that the article lacks reflection on the effects of lawyers using mimicry in their interactions with clients. Can a profession based on social trust employ social influence tools to manipulate their clients? It seems that this issue is not given sufficient attention in the article.

- Done! We fully agree with your great point! Please see p.23 - 24.

Despite the mentioned faults, I find the submitted article interesting and believe it is worth publishing in PlosOne after implementing the necessary corrections.

***

- Finally, once again we would like to express our gratitude to the Reviewers for their input and insight which - we hope - improved our manuscript.

6. PLOS authors have the option to publish the peer review history of their article (what does this mean?). If published, this will include your full peer review and any attached files.

Do you want your identity to be public for this peer review? For information about this choice, including consent withdrawal, please see our Privacy Policy.

Reviewer #1: No

Reviewer #2: No

---

## [Decision Letter · Decision Letter 1]

27 Sep 2023

Mimicry and law: Experiments in a natural setting of a law company

PONE-D-23-10321R1

Dear Dr. Muniak,

We’re pleased to inform you that your manuscript has been judged scientifically suitable for publication and will be formally accepted for publication once it meets all outstanding technical requirements.

Kind regards,

Sylvester Chidi Chima, M.D., L.L.M.

Academic Editor

PLOS ONE

Additional Editor Comments (optional):

Reviewers' comments:

Reviewer's Responses to Questions

**Comments to the Author**

1. If the authors have adequately addressed your comments raised in a previous round of review and you feel that this manuscript is now acceptable for publication, you may indicate that here to bypass the “Comments to the Author” section, enter your conflict of interest statement in the “Confidential to Editor” section, and submit your "Accept" recommendation.

Reviewer #1: All comments have been addressed

Reviewer #2: All comments have been addressed

2. Is the manuscript technically sound, and do the data support the conclusions?

Reviewer #1: Yes

Reviewer #2: Yes

3. Has the statistical analysis been performed appropriately and rigorously? 

Reviewer #1: Yes

Reviewer #2: Yes

4. Have the authors made all data underlying the findings in their manuscript fully available?

Reviewer #1: Yes

Reviewer #2: Yes

5. Is the manuscript presented in an intelligible fashion and written in standard English?

Reviewer #1: Yes

Reviewer #2: Yes

6. Review Comments to the Author

Reviewer #1: thank you for the revision. I don't have additional comments whatsoever, just typing to meet the required minimum character count.

Reviewer #2: With great satisfaction, I have read the revised version of the article. In my opinion, the authors have addressed all the critical comments and significantly improved both the structure and content of the article. I am fully satisfied with the changes made – I am convinced that the article can be published in this form in PLOS One.

7. PLOS authors have the option to publish the peer review history of their article (what does this mean?). If published, this will include your full peer review and any attached files.

Reviewer #1: No

Reviewer #2: No

---

## [Editor Report · Acceptance letter]

5 Oct 2023

PONE-D-23-10321R1 

Mimicry and law: Experiments in a natural setting of a law company 

Dear Dr. Muniak:

I'm pleased to inform you that your manuscript has been deemed suitable for publication in PLOS ONE. Congratulations! Your manuscript is now with our production department. 

Kind regards, 

on behalf of

Professor Sylvester Chidi Chima 

Academic Editor

PLOS ONE